# Human *Schistosoma* exposure risk in rice fields and an exploration of fish species for snail and schistosomiasis biocontrol

Alexandra Sack[1,2,3☉], Emily Selland[1,2☉], Sidy Bakhoum[1,2,4], Momy Seck[5], Nicolas Jouanard[5], Louis Dossou Magblenou[5], Jason R. Rohr[1,2,6]*

1 Department of Biological Sciences, University of Notre Dame, Notre Dame, Indiana, United States of America, 2 Eck Institute of Global Health, University of Notre Dame, Notre Dame, Indiana, United States of America, 3 The Carter Center, Atlanta, Georgia, United States of America, 4 Department of Animal Biology, University Cheikh Anta Diop, Dakar, Senegal, 5 Station d'Innovation Aquacole, Saint-Louis, Senegal, 6 Environmental Change Initiative, University of Notre Dame, Notre Dame, Indiana, United States of America

☉ These authors contributed equally to this work.
* jrohr2@nd.edu

## Abstract

Schistosomiasis is a devastating parasitic disease in which the infectious stage to humans is released by intermediate host snails. The Senegal River Basin (SRB) is a high-risk area for both urogenital and fecal human schistosomiasis and has extensive rice cultivation. However, occupational risk of schistosomiasis to people working in irrigated rice fields is not well established. We performed intermediate host snail surveys from 2022-2023 in rice fields and irrigation canals throughout the SRB. We discovered human schistosome-shedding snails in rice fields and adjacent irrigation canals during the rice growing and non-growing seasons, establishing a clear occupational exposure risk to rice farmers. Relative to the non-growing season, this risk was higher in the rice growing and harvest season when more people are in the rice fields. Rice-fish co-culturing might reduce this occupational risk to rice farmers if local fish species consume enough snail intermediate hosts to reduce *Schistosoma* transmission. Our predation trials revealed that local *Heterotis niloticus* and *Hemichromis* spp. fish consumed significant numbers of *Biomphalaria pfeifferi* and *Bulinus* spp. snails, and separate trials revealed that these same snail species exhibited only moderate avoidance and refuge use responses to fish chemical cues. These results indicate that there is exposure to *Schistosoma* parasites in rice fields in the SRB and introducing local fish to rice fields has promise for reducing this exposure as well as providing a protein source to rice farming families. We encourage future studies to more fully explore the benefits of rice-fish co-culturing in the West Africa.

**Data availability statement:** Data is available at https://figshare.com/articles/dataset/Data_for_Field_Surveys_and_Experiments-Sack_Selland_xlsm/28861280?file=53963564

**Funding:** This work was supported by funds from the National Science Foundation (DEB-2017785 to JRR; DEB-2109293 to JRR; BCS-2307944 to JRR; ITE- 2333795 to JRR; 2236418-002 to EKS), Frontiers Planet Prize (to JRR), and the University of Notre Dame Poverty Initiative (to JRR). The funders had no role in study design, data collection and analysis, decision to publish, or preparation of the manuscript.

**Competing interests:** The authors have declared that no competing interests exist.

## Introduction

Schistosomiasis, a debilitating disease caused by trematodes, is the second most devastating parasitic disease worldwide [1]. Children with chronic infections can develop anemia, malnutrition, and learning deficiencies. Over time, the parasites can damage the liver, lungs, and bladder and even cause kidney failure [2]. The causative agents of fecal and urogenital schistosomiasis, *Schistosoma mansoni* and *Schistosoma haematobium,* are spread by their intermediate snail hosts *Biomphalaria* spp. and *Bulinus* spp., respectively, and account for almost all human infections in Africa [3]. While the drug praziquantel is relatively effective at treating schistosomiasis, this disease continues to defy control efforts because people quickly get re-infected when they return to water sources with infected snails [4]. For this reason, the World Health Organization (WHO) recommends the implementation of snail control alongside mass drug administration to control schistosomiasis [5].

Given that contact with fresh water, such as while bathing, drawing water, and washing, is often the source of *Schistosoma* infections, there might be an occupational exposure risk for agricultural laborers who have extended contact with surface water, such as rice farmers. Rice is grown in standing water that might harbor freshwater snails shedding free-swimming cercaria, the stage of the parasite that infects humans [5]. However, an occupational risk is not well established in the literature, and the last systematic review of schistosomiasis to include rice fields in 2006 only included four studies [6]. The Senegal River Basin (SRB) is a prime location for evaluating an occupational risk of schistosomiasis for rice farmers because it has a high prevalence of rice cultivation and urogenital and fecal schistosomiasis [3]. Despite widespread rice and schistosomiasis, the importance of rice cultivation to the transmission of human *Schistosoma* parasites in the region remains unknown because rice fields in the area are often drained before harvesting and can remain unflooded in the non-growing season [6], which might reduce snail survival and thus schistosomiasis risk. Hence, a better understanding of which water bodies represent risks of *Schistosoma* transmission is needed to design effective schistosomiasis control.

If rice farming represents an occupational risk of schistosomiasis, snail control using molluscicides would be problematic in rice fields because molluscicides are often toxic to humans and wildlife [4, 7], and thus, alternative approaches to snail control would likely be necessary in rice fields. One viable approach suggested by researchers and the WHO is the introduction of snail predators, such as fish or crayfish [8]. Fish have been used to control non-*Schistosoma*-transmitting snail species that are pests in rice fields in Asia as part of rice-fish co-culturing systems. This practice, however, has not been tested as a control for *Schistosoma*-transmitting snails. The efficacy of any biocontrol mechanism for schistosomiasis can be complex, often depending on the foraging and competitive abilities of the biocontrol agent and the anti-predator or anti-competitor responses of the snails, among other things [9, 10]. For example, snails can detect predator chemical cues (kairomones) [11] and respond with antipredator behaviors, such as predator avoidance and refuge use, and some snails can even display predator-species-specific responses

[12, 13]. Predator avoidance often entails moving away from high-risk locations and moving towards or above the water line, whereas refuge use often involves moving into dense vegetation.

In addition to potentially reducing the risk of schistosomiasis, rice-fish co-culturing might provide other important benefits, such as improving food security. Where practiced globally, rice-fish requires lower inputs for both the fish and the rice relative to raising each alone [14]. Fish provide food to farmers while also reducing the number of macroinvertebrates that serve as pests. The fish also help make carbon, nitrogen, and phosphorous more accessible, increasing rice productivity [15, 16]. In doing so, the fish introductions reduce the amount of fertilizer and pesticide needed in rice cultivation, which is both environmentally and economically beneficial [14].

A review of fish native to the SRB revealed several species that might depredate or compete with *Schistosoma*-host snails, which could reduce snail populations or their food source, and thus, their ability to shed *Schistosoma* cercariae [17]. Of particular interest in the SRB are four native fish species that have been used in previous aquaculture rice field systems [18–22]: North African catfish (*Clarias gariepinus)*, Nile Tilapia (*Oreochromis niloticus*), Jewel Cichlids (*Hemichromis* spp.), and African Bonytongue (*Heterotis niloticus*). While all four of these species can serve as food for farmers or as an income source, *Heterotis niloticus* has been of special interest to the greater aquaculture community because of its high value as a food item and its omnivorous diet, which includes snails [23, 24]. If these native fish species consume enough snail intermediate hosts to reduce transmission, then rice-fish co-culturing might reduce any occupational risk of schistosomiasis to rice farmers. However, there is almost nothing known about the potential of these species to reduce snail populations in rice fields.

In this study, we first assessed whether there is an occupational *Schistosoma* exposure risk for rice farmers by performing surveillance of rice fields and irrigation canals near rice in the SRB across the rice growing and non-growing seasons. This allowed us to test whether there were intermediate host snails shedding *Schistosoma* cercariae in these fields at these times. Next, we conducted a series of experiments to determine whether the four focal fish species depredate *Biomphalaria pfeifferi* and *Bulinus* spp.*, the intermediate hosts of *S. mansoni* and *S. haematobium* respectively, as well as snails of genus *Lymnaea*. *Lymnaea* snails were included because they are hosts of *Fasciola* spp. trematodes that cause cattle sickness and mortality in this region, and rice straw is a commonly reported fodder for cattle. We also tested whether these same snail species exhibit any antipredator behaviors in response to the chemical cues of these four native fish species to help understand whether the foraging trials were a product of just the foraging efficiencies of the fish or also antipredator behaviors of the snails. We conducted these studies in an effort to guide rice-fish co-culturing to improve food production and hopefully also reduce any occupational risk of schistosomiasis.

## Methods

### Site selection

Rice fields in the SRB are all irrigated by canals from local rivers. Rice fields were drawn from four areas in both the Saint-Louis and Richard Toll regions. Historically, the SRB had two crops of rice per a year (dry and wet season). However, during our study, there was a shift towards more vegetable production during the wet season and rice during the dry season [25]. Most of the fields in this study were used as onion fields during the off season and thus were not flooded. However, following the study, many farmers have returned to two rice crops a year, and the Senegalese government supports increased rice production.

### Field sampling method

Owners of rice fields gave verbal permission for sampling and permission for testing in communal water sources, such as certain canals, was given verbally by village leadership when applicable. Land was privately owned, and no protected species were sampled. Water level and temperature in the fields and canals were measured at sampling points. Snail sampling was done with a combination of sweep net surveys and snail traps with permission from local farmers. Sweep

netting was used in irrigation canals and in areas of rice fields that would not damage rice, such as the canal entrance, in bare spots, non-rice vegetation, and along the sides of the field. We designed snail traps using recycled plastic water bottles. These traps were baited with mango, which was effective at attracting snails in both the laboratory and rice fields [26]. Snail traps were placed on each side of the field and one at the canal entrance (i.e., five traps/field) and checked the next day (i.e., 18–24 hours later). Snails were counted and identified to species or species complex in the field, and *Biomphalaria pfeifferi. Bulinus globosus/truncatus*, and *Bulinus forskalii/senegalenesis* were taken to the laboratory. In the laboratory, we sized the snails using a digital photograph and ImageJ software, exposed them to light to stimulate cercarial production, and recorded the number of cercariae shed over a 1.5 hour period, utilizing standardized staining and microscopy techniques [27]. Aquatic vegetation was identified on-site in both canals and rice fields with help from local field team members and classified as either submergent or emergent vegetation. Snail traps were only used in the summer of 2022 during the rice harvest season. Sampling was repeated three times in the planting/harvesting season (summer) of 2022, non-growing season (March) 2023, and planting/harvesting season (summer) of 2023. Total snails and percent shedding human schistosomes was calculated for each period, pooled across traps and sweeps.

We conducted a generalized linear mixed effects model with snail abundance (only *Bulinus globosus/truncatus* and *Biomphalaria pfeifferi*) per sweep as our a negative binomially distributed response variable (lme4 package, glmer.nb function); location (field or canal), presence of submergent vegetation, presence of emergent vegetation and season as independent variables; and field as a random intercept nested within rice region. Prevalence of shedding snails was not used as an outcome because of the small sample size of snails that shed cercariae.

## Set-up and animal care for laboratory experiments

Laboratory experiments were conducted to test whether snails would avoid fish predators and whether native fish species would consume snails collected from local water bodies. *Bulinus globosus/truncatus, Biomphalaria pfeifferi,* and *Lymnaea* spp. were collected in the same week as each temporal block of the experiment. Snails were housed in 5-L freshwater holding tanks before and after trials and were provided with *Ceratophyllum demersum* from waterbodies, which had algae for the snails to eat. Previous research showed that this is the preferred aquatic vegetation for these snails in the SRB [27].

The four native fish species included were North African catfish (*Clarias gariepinus),* Jewel cichlids (*Hemichromis* spp.), African Bonytongue (*Heterotis niloticus*), and Nile tilapia (*Oreochromis niloticus*). Fish species were held in their home tanks, which were large freshwater, aerated, solid-sided holding tanks, before and after experiments at the following densities: 50 individuals each per 1 m$^3$ tank for *C. gariepinus* and *Hemichromis* spp., 20 individuals per 0.5 m$^3$ tank for *H. niloticus*, and 500 individuals per 30 m$^3$ tank for *O. niloticus*. Fish ranged in size within each tank, but *H. niloticus* were the largest and *Hemichromis* spp. the smallest. All fish were fed their typical food (a mix of fishmeal, peanut cake, rice bran, broken corn, mineral premix, vitamin premix, fish oil, and water softener), including the day before the predation experiment. The holding tanks for these fish provided the water for the predator avoidance experiment. All experimental trials were performed at Station d'Innovation Aquacole (SIA) in Saint-Louis, Senegal and the water from the facility was used for all experiments. Notre Dame's IACUC approved this protocol (ID: 23-01-7615, PI: Rohr), and no permits were required in Senegal as we were working within an established aquaculture facility. All statistical tests were performed using R studio (v. 1.4.1717).

## Experiment 1: Predator avoidance behavior

In Experiment 1, we compared the predator avoidance response of three different genera of snails (*Bulinus globosus/truncatus, Biomphalaria pfeifferi*, *Lymnaea* spp.) to fish chemical cues. Groups of 30 snails (10 from each genus) were placed overnight in a 25-L tank with or without 30 g of aquatic vegetation (*C. demersum*). The *C. demersum* vegetation was collected within 48 hours of the start of the experiment, and all snails and other invertebrates were picked off from the vegetation. To ensure consistent sizes across treatments, snails were sized on a grid before being placed in the tank for an

overnight acclimation period. Average snail size did not differ among the tanks (ANOVA: $F = 0.21$, $p = 0.65$). The following morning, the position of the snails in the tank was recorded as bottom, side below waterline, at waterline, above waterline, or in vegetation (for vegetation tanks). Every two hours for 10 hours, 500 mL of water from the stock tank of one of the four fish species (African catfish, Jewel cichlids, African Bonytongue, and Nile tilapia) was then gently added to the same side of the tank to minimize disruption to the surface water and snail locations. A single stock tank was used for all replicates for each fish species. For the control replicates, clean water was added. Snail locations were recorded one hour after each chemical cue addition. Each fish-by-vegetation treatment (presence/absence) was replicated across three temporal blocks. Avoidance is defined as hiding in vegetation when available and being at the waterline or exiting the water. Since exiting the water would lead to desiccation in a very short time in Senegal, at the water line was considered predator avoidance behavior, as snails had greater access to leave without the constant threat of desiccation. These two behaviors were tested separately, as only a subset of tanks had access to vegetation. Snails were returned to their holding tank after the experiment concluded. We used a mixed effects model with proportion of total snails showing avoidance behavior as the dependent variable (i.e., a binomial error distribution), fish cue (four fish plus a control), snail species, vegetation treatment, and hour of the trial as crossed independent variables, block as a fixed main effect, and tank as a random intercept. We could only analyze the proportion of snails in the vegetation for the half with vegetation. Hour was included in the models to test if the snails became habituated to the chemical signals. We calculated pairwise comparisons of estimated marginal means (*emmeans* in R) to compute contrasts across categorical predictors.

## Experiment 2: Selective predation

In Experiment 2, we compared 24-hour predation by native fish on the same three genera of snails in tanks with and without aquatic vegetation. Due to space limitations, this experiment was conducted with half the replicates for each treatment occurring in each of two time blocks. Experiments lasted for 24 hours, and *C. gariepinus* and *O. niloticus* were only included in the first round because of the stress levels of the fish. This experiment was performed on three groups of 15 snails (5 of each genus) placed overnight in a 25-L tank with or without 60 g of aquatic vegetation (*C. demersum*). The *C. demersum* vegetation was collected within 48 hours of the start of the experiment, and all snails and other invertebrates were removed from the vegetation. Snails were sized before being placed in the tank, allowing us to test whether fish prefer to eat snails of particular sizes. The following morning, one fish from each of four different species was weighed, had its length measured, and was added to the tank. The four fish species were the same as used in Experiment 1. For control tanks, no fish were added. The average number of live snails were recorded every 2 hours for 10 hours, and after 24 hours. Fish were returned to their home tank if they displayed signs of distress or illness. Due to signs of stress from *C. gariepinus* and *O. niloticus*, they were excluded from the second temporal block. Thus, the second block included *Hemichromis* spp., ($n = 8$) in the 25-L tanks and *H. niloticus* ($n = 4$) moved to 200-L tanks. In the second block, snail survival was checked only every 6 hours to reduce stress on the fish, with a 12-hour break overnight. At the end of the study, any remaining snails were sized as described above. Living snails and all fish were returned to their respective tanks after the experiment concluded. Our base and most complex statistical model was a binomial regression with the number of dead snails in each tank of each species as the dependent variable, snail species, fish species, and the presence of aquatic vegetation as crossed independent variables, and tank and block included as random intercepts. We considered all reduced, nested models to identify the most parsimonious model with the lowest corrected Akaike Information Criteria (AICc).

To test for size-specific predation rates on the snails, we conducted a before-after control-impact analysis with the mean size of surviving snails before and after the trial in each tank as the Gaussian distributed dependent variable; main effects and interactions among fish species (only *Hemichromis* spp. and *H. niloticus*), time (Before versus After), and snail species as independent variables; and accounting tank and block as random intercepts. We compared three models to identify the most parsimonious model with the lowest AICc: one with a 3-way interaction between fish species, time, and

snail species, one with a 2-way interaction between fish species and time with an additive effect of snail species, and a third model with additive effects of fish species, time, and snail species.

## Results

### Field sampling

Water depth and temperature varied between irrigation canals and fields. Rice fields were shallower than irrigation canals; depth at sampling points in irrigation canals and rice fields averaged 30 cm (range: 2–125 cm) and 12 cm (range: 0.25 cm to 33 cm), respectively. At least partly due to the shallower depth, water in rice fields (mean: 24.9 °C, range: 22.2-30 °C) was warmer than irrigation canals (mean: 23.1 °C, range: 19.5-30 °C). During sampling of rice fields and canals, ten vegetation species were recorded, and grouped as submerged or emergent.

During the four rounds of sampling in 2022 and 2023, while *Biomphalaria* snails were only occasionally found, *Bulinus* snails were found in most rice fields and rice irrigation canals in the growing and non-growing seasons (Table 1, Fig 1). Additionally, snails were regularly shedding human *Schistosoma* cercariae in rice fields and rice irrigation canals. There was no difference in the number of snails that were collected with traps or sweep nets ($Z = -1.193$, $p = 0.233$); thus, sampling method was not included in snail abundance models. Snails tended to be more abundant in irrigation canals than rice fields, although this effect was not significant ($Est = 0.68$, $SE = 0.37$, $p = 0.066$; Fig 2). Snails were positively associated with both submergent ($Est = 1.86$, $SE = 0.35$, $p < 0.001$) and emergent vegetation ($Est = 0.83$, $SE = 0.33$, $p = 0.012$), although the highest predicted abundance of snails per sweep occurred for sweeps in irrigation canals where submergent vegetation was present (Fig 2A). *Bulinus globosus/truncatus* and *B. pfeifferi* snails were more abundant in both fields and

**Table 1. Results of sampling across all seasons and study sites reflecting the number of fields and canals sampled during the season, the number of fields and canals where snails were found, and the total snails captured at the species-group level. The mean and standard deviation of snails captured per site and the prevalence of snails shedding human-infectious *Schistosoma* cercariae are also presented at the species-group level. The mean and standard deviation are calculated over number of sites where the species was identified because the sampling methodology different between seasons and between canals and fields (see Methods). *Schistosoma* cercariae were identified morphologically. The temperature of water is reflected as a range encompassing canal and field data.**

| Dependent variables | June 2022 | | March 2023 | | July 2023 | |
|---|---|---|---|---|---|---|
| | Growing/Harvesting | | Non-growing | | Growing/Harvesting | |
| | Canal | Field | Canal | Field | Canal | Field |
| Sites with snails | | | | | | |
| Bi. pfeifferi | 1/12 | 0/11 | 3/6 | 0/1 | 6/9 | 4/4 |
| Bu. globosus/truncatus | 9/12 | 6/11 | 4/6 | 1/1 | 7/9 | 3/4 |
| Bu. forskalii/sengalensis | 6/12 | 6/11 | 1/6 | 0/1 | 1/9 | 1/4 |
| Total snails captured* | | | | | | |
| Bi. pfeifferi | 1 (1) | 0 | 105 (30-44) | 0 | 192 (5-95) | 79 (1-63) |
| Bu. globosus/truncatus | 375 (1-316) | 87 (2-54) | 105 (3-78) | 1 (1) | 154 (2-72) | 28 (1-14) |
| Bu. forskalii/sengalensis | 53 (1-27) | 344 (10-186) | 1 (1) | 0 | 16 (16) | 2 (2) |
| Prevalence of shed *Schistosoma* cercariae | | | | | | |
| Bi. pfeifferi | 0 | – | 12.38% | – | 4.69% | 3.80% |
| Bu. globosus/truncatus | 3.73% | 1.15% | 22.88% | 0 | 3.25% | 14.29% |
| Bu. forskalii/sengalensis | 0 | 0.29% | 0 | – | 6.25% | 0 |
| Temp. range (C) | 24.1 – 27.9 | | 23.7 – 26.7 | | 26.1 – 30.1 | |

*values in parentheses represent the range of snails captured per site that the species was identified at. If only one number if present, then that species of snail was only captured at one of the sampled sites during that time period.

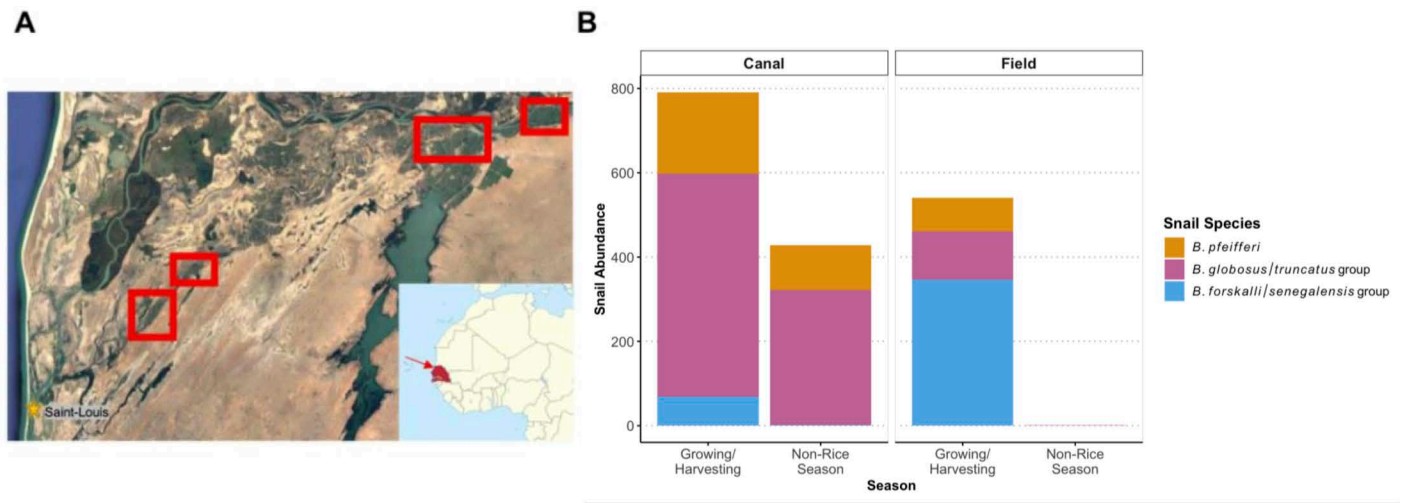

**Fig 1. (A) General area of rice fields sampled.** (B) Total snail abundance sampled in rice fields and canals across the four regions.

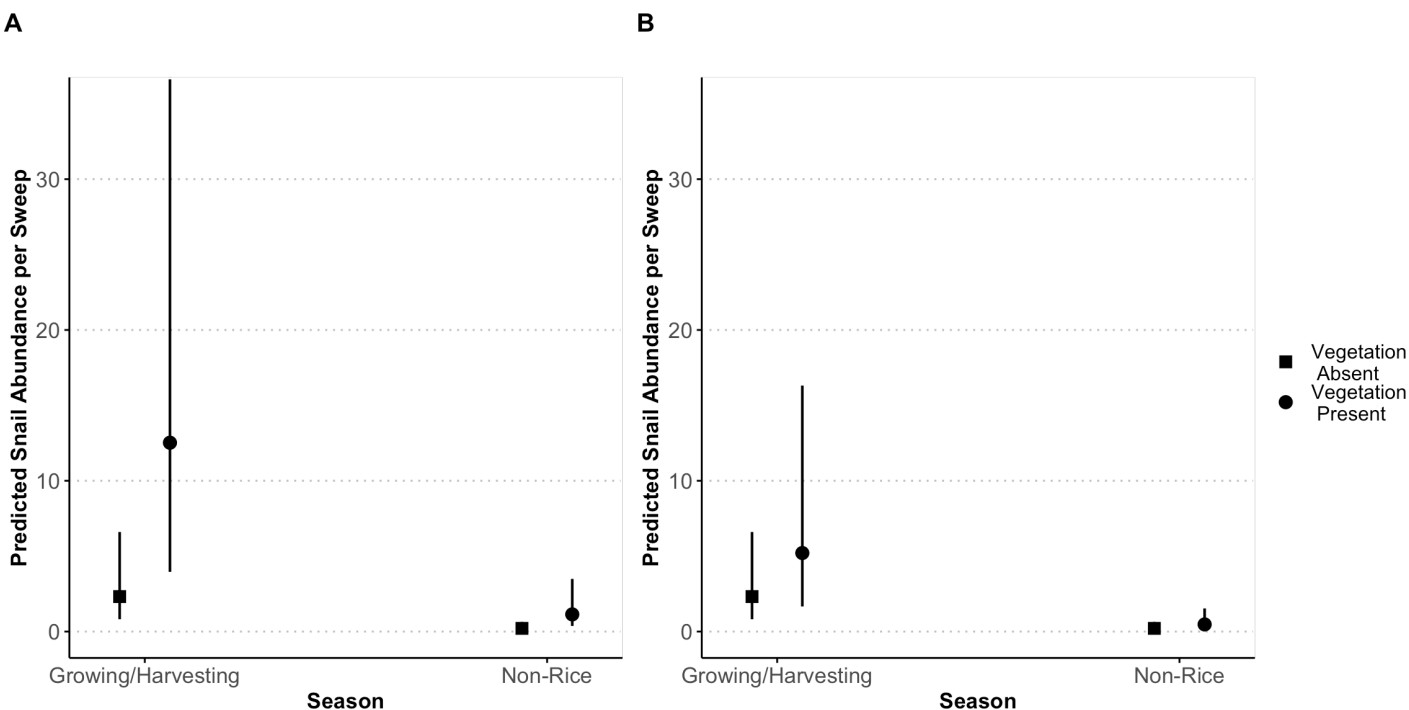

**Fig 2. Snail abundance per sweep across non-rice and rice growing/harvesting seasons.** Dots represent predicted mean values (and 95% confidence intervals) of snail abundance per sample (trap or sweep) for seasons across the presence and absence of (**A**) submergent and (**B**) emergent vegetation from a generalized linear mixed model with a negative binomial distribution of combined *B. pfeifferi* and *B. globosus/truncatus* abundance.

canals during the rice growing season compared to the non-growing season (*Est* = 2.06, *SE* = 0.36, *p* < 0.001; Fig 2). As a reminder, snails were not found in rice fields in the non-growing season because the fields were drained and dry (Fig 1).

## Experiment 1: Predator avoidance behavior

Of the 30,000 snail hours observed (i.e., 300 snails x 100 h of observations each), only in 21 hours (0.7%) were snails observed above the waterline. Of the 21 hours, 15 hours were by *Lymnaea spp.,* and only 6 of them were in tanks with vegetation; a single *Lymnaea* snail died from being out of the water. In contrast, snails were observed at the waterline for 877 hours, and snails were in the vegetation for 1,937 hours. The remaining hours were not near a refuge. For avoidance behavior at or above the waterline, the best model included the fixed effect of time of the observation in the trial and the three-way interaction of fish chemical cue-by-snail species-by-vegetation presence (*p* < 0.005; Table 2, Fig 3A; delta AICc > 4 from next best model). Snails were less likely to be above the water line when vegetation was present (main effect *p* < 0.001) and later in the trials (Est = -0.039, SE = 0.015, *p* = 0.01; Fig 3A). When vegetation was absent, *B. pfeifferi* were more likely to be above the water line than *Bulinus globosus/truncatus* in control tanks (*p* = 0.02) and *Lymnaea* spp. in tanks with *O. niloticus* chemical cues (*p* = 0.04) but were less likely to be above the water line than *Lymnaea* spp. in tanks with *C. gariepinus* cues (*p* = 0.04; Fig 3A). When vegetation was present, both *B. pfeifferi* and *Bulinus globosus/truncatus* were less likely to be above the water line than *Lymnaea* spp. in tanks with *O. niloticus* cues (*p* = 0.002 and *p* = 0.02, respectively; Fig 3A). This same pattern in behavior for both *B. pfeifferi* and *Bulinus globosus/truncatus* was seen in tanks with *C. gariepinus* and *Hemichromis* spp. cues, although it was marginally nonsignificant (*p* = 0.06 for all; Fig 3A). Pairwise comparisons across fish species chemical cues by snail species and vegetation presence showed no significant pairwise differences (S1 Table). There was no significant difference in starting snail size across the different treatment arms (*F* = 0.21, *p* = 0.65).

For snails in tanks with vegetation, the best model included the two-way interaction between fish species chemical cues and snail species (*p* = 0.006; Fig 3B), and the two-way interaction between fish species chemical cues and time of the observation during the trial (*p* = 0.04; Fig 3C). Although, the delta AICc was less than 2 between this model and the model that only included the interaction between fish species chemical cues and snail species. No pairwise comparisons across

**Table 2. Analysis of variance table for the best binomial model (selected based on lowest AICc) of proportion of snails showing predator avoidance behavior in response to chemical cues by different fish species, snail species tested, vegetation presence, and time during the trial. Avoidance behavior is defined as moving above or being at the waterline (when aquatic vegetation is present or absence) or hiding in vegetation (for the subset of boxes with vegetation). Tank and temporal block were included as random intercepts.**

| Avoidance Behavior | Variable | Chi Square | p-value |
|---|---|---|---|
| Above or at water line | Fish chemical cues | 4.29 | 0.37 |
| | Snail Species | 4.13 | 0.13 |
| | Aquatic Vegetation | 44.13 | <0.0001 |
| | Fish Cues * Snail Species | 20.79 | 0.008 |
| | Fish Cues * Vegetation | 3.64 | 0.46 |
| | Snail Species * Vegetation | 14.96 | <0.001 |
| | Fish Cues * Snail Species * Vegetation | 25.35 | <0.005 |
| | Hour since start | 6.57 | 0.01 |
| In Vegetation | Fish chemical cues | 8.58 | 0.072 |
| | Snail Species | 20.75 | <0.0001 |
| | Hour since start | 0.17 | 0.68 |
| | Fish Cues * Snail Species | 21.39 | 0.0062 |
| | Fish Cues * Hour | 10.19 | 0.037 |

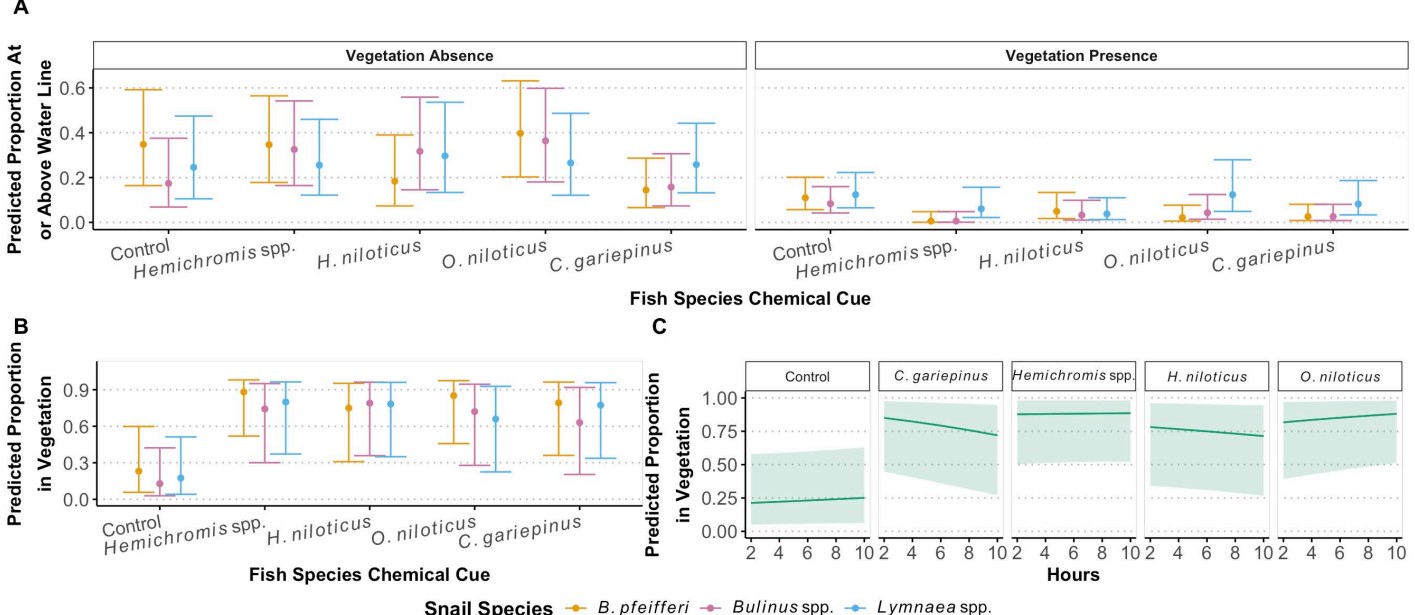

**Fig 3. (A) Predicted probabilities of snails at or above the water line across different fish species chemical cues.** Dots represent predicted mean (and associated 95% confidence intervals) proportion of the snails at or above the water line after the addition of the fish chemical cue from a generalized linear model with a binomial error distribution across snail species. (B) Predicted probabilities of snails in vegetation in the presence of fish species chemical cues from the interaction of snail species and fish species. Dots represent predicted mean (and associated 95% confidence intervals) proportion of the snails in vegetation after the addition of the fish chemical cues from a generalized linear model with a binomial error distribution across snail species. (C) Predicted probabilities of snails in vegetation across hours of experiment after exposure to fish chemical cues. Snails exposed to *C. gariepinus* chemical cues decreased their use of the vegetation over time as compared to snails in the control tanks, but the proportion of snails in the vegetation through time otherwise did not significantly differ from the controls for any other fish species. Lines represent the predicted proportion of snails in the vegetation across time for each fish species and the shaded region represents the associated 95% confidence band.

fish and snail species were significantly different from one other, and, averaging over snail species, snails were not significantly more likely to be in vegetation when exposed to fish cues compared to the control. However, *B. pfeifferi* tended to be in vegetation more when exposed to *Hemichromis* spp. cues compared to the control ($p = 0.088$; Fig 3B) and *Bulinus globosus/truncatus* were marginally more likely to be in vegetation when exposed to *H. niloticus* cues compared to the control ($p = 0.0799$; Fig 3B). *Lymnaea* spp. were not any more likely to be in the vegetation for any fish species compared to the control. Trends for the proportion of snails in vegetation across time varied only between the control and *C. gariepinus* chemical cues ($p = 0.02$; Fig 3C), where the proportion of snails exposed to *C. gariepinus* chemical cues in vegetation decreased over time.

## Experiment 2: Selective predation

For the four species used in this experiment, average weight and length were 89.2 g (SE = 17.3) and 22.0 cm (SE = 2.1) for *C. gariepinus,* 10.0 g (SE = 1.0) and 7.4 cm (SE = 0.4) for *Hemichromis* spp., 253.1 g (SE = 20.2) and 29.4 cm (SE = 0.75) for *H. niloticus,* and 72.7 g (SE = 12.4) and 17.7 cm (SE = 1.5) for *O. niloticus.* The final model for predation of snails accounted for both fish species and snail species as main effects (AICc = 173.06), as compared to a model with their interaction term (AICc = 186.53) or the inclusion of aquatic vegetation (AICc = 175.29). There was a significant effect of snail species ($p = 0.006$; $\chi^2 = 1.88$) and of fish species ($p = 0.046$; $\chi^2 = 9.67$) on the proportion of dead snails after 24 hours. After accounting for both tank and block as random intercepts and snail species, snails exposed to *Hemichromis* spp. had 1.15 odds of dying compared to the control tank ($p = 0.03$; Table 3, Fig 4B). *Bulinus globosus/truncatus* had 1.27 odds of dying

**Table 3. Odds ratio of pairwise contrasts from the binomial model of proportion of snails depredated 24 hours after the start of the trial by fish species. Tank and temporal block were included as random intercepts.**

| Pairwise comparison | Contrast | Odds Ratio | SE | p-value |
|---|---|---|---|---|
| Snail Species (results are averaged over levels of Fish Species) | *Bi. pfeifferi – B. globosus/truncatus* | 0.27 | 0.116 | 0.0068 |
| | *Bi. pfeifferi – Lymnaea* spp. | 0.539 | 0.251 | 0.38 |
| | *B. globosus/truncatus – Lymnaea* spp. | 1.998 | 0.727 | 0.14 |
| Fish Species (results are averaged over levels of Snail Species) | Control – *Hemichromis* spp. | 0.15 | 0.097 | 0.029 |
| | Control – *H. niloticus* | 0.27 | 0.19 | 0.34 |
| | Control – *O. niloticus* | 0.29 | 0.24 | 0.57 |
| | Control – *C. gariepinus* | 0.3 | 0.24 | 0.58 |
| | *Hemichromis* spp. – *H. niloticus* | 1.81 | 0.78 | 0.65 |
| | *Hemichromis* spp. – *O. niloticus* | 1.96 | 1.21 | 0.81 |
| | *Hemichromis* spp. – *C. gariepinus* | 1.97 | 1.21 | 0.81 |
| | *H. niloticus – O. niloticus* | 1.08 | 0.73 | 1.00 |
| | *H. niloticus – C. gariepinus* | 1.09 | 0.73 | 1.00 |
| | *O. niloticus – C. gariepinus* | 1.01 | 0.80 | 1.00 |

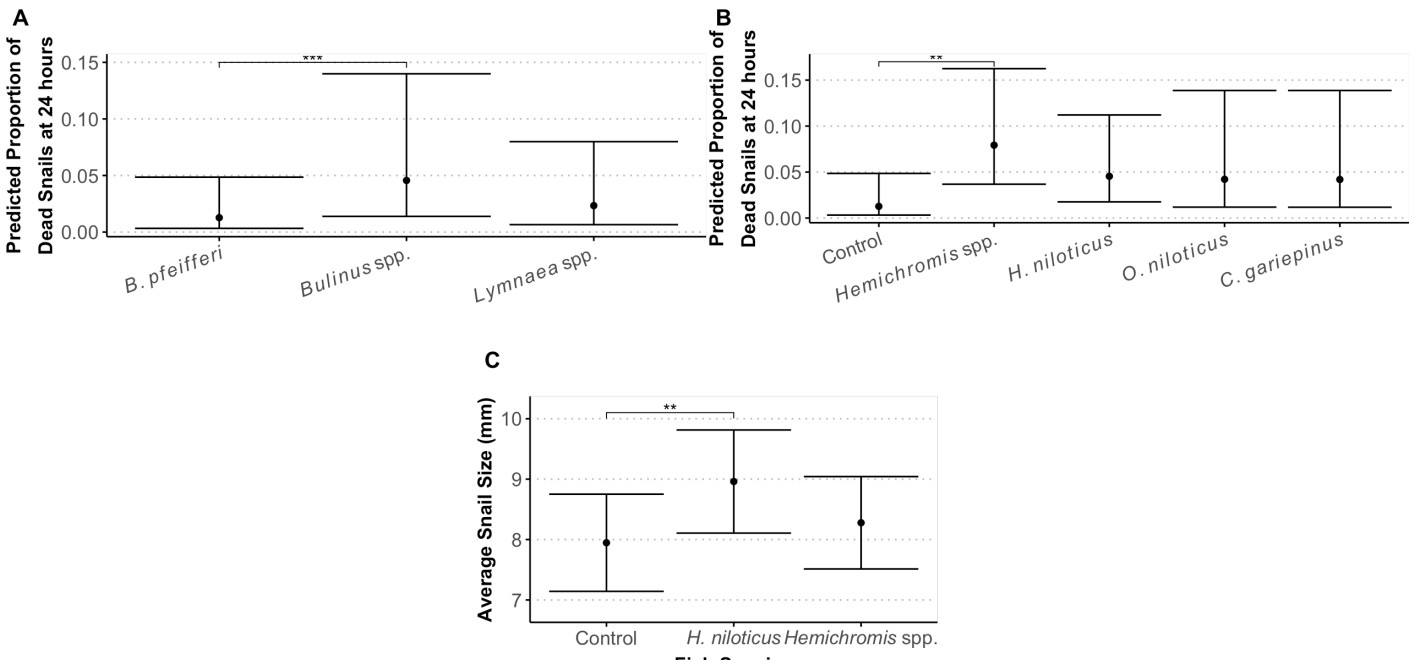

**Fig 4. (A. B) Predicted proportion of snails dead after the 24-hour predation experiment.** The proportion of dead snails after 24 hours varied both by (A) snail species and by (B) fish species. (A) *Bulinus* spp. were more likely than *B. pfeifferi* to be depredated in the predation experiment, controlling for the fish species. (B) Snails exposed to *Hemichromis* spp. were more likely to be consumed than those in the control tanks, indicating that *Hemichromis* spp. effectively predated the snail species. Shown are means and 95% confidence intervals. (C) Mean and standard deviation of snail (*Lymnaea* spp, *B. pfeifferi*, and *Bulinus globosus/truncatus*) size across time exposure to fish predators. * = *p*-value ≤0.05 and ≥0.01, *** = *p*-value <0.01 and ≥0.001.

as compared to *B. pfeifferi* (*p* = 0.007; Table 3, Fig 4A) but were not significantly more likely to die compared to *Lymnaea* spp. (*p* = 0.14; Table 3, Fig 4A). The proportion of *B. pfeifferi* and *Lymnaea* spp. dead after 24 hours were not significantly different from each other (*p* = 0.38). *Hemichromis* spp. and *C. gariepinus* were observed to remove snails from their shells and visible marks on the outside of the shell were often left by this process.

For the size of snails before and after predation, the best model included only the additive effects of snail species (*p* < 0.0001; $x^2$ = 713.33), fish species (*p* = 0.02; $x^2$ = 8.27), and time (*p* = 0.03; $x^2$ = 4.76). The species of snails were different sizes: *Lymnaea* spp. were an average of 10.53 mm, which was significantly larger than both *B. pfeifferi* (average size of 8.27 mm, *p* < 0.0001) and *Bulinus globosus/truncatus* (average size of 6.86 mm, *p* < 0.0001). Snails placed with *H. niloticus* were larger at the start of the experiment than those in the control tanks (*p* = 0.027; Fig 4C), but snails in tanks with *Hemichromis* spp. did not differ in size compared to those in controls (*p* = 0.51; Fig 4C) or *H. niloticus* tanks (*p* = 0.14; Fig 4C). In the model including the interaction between fish species and time, the interaction was not significant (*p* = 0.37; $x^2$ = 1.98). However, surviving snails exposed to the smallest of the four fish predators, *Hemichromis* spp., were larger after predation than before (*Est* = 0.41, *p* = 0.01; S1 Fig).

## Discussion

Schistosomiasis is a devastating disease that, despite treatment with drugs, has high reinfection rates when infected snails persist in water sources with which people have contact [4]. Our study (1) demonstrates a schistosomiasis occupational risk to rice farmers from both rice fields and nearby irrigation canals and (2) identifies fish species native to Senegal that readily consume intermediate host snails, thus representing candidates for biocontrol of schistosomiasis in rice fields and possibly other settings.

During the rice growing seasons of 2022 and 2023, intermediate host snails were found in most fields and irrigation canals with many fields also having snails shedding *Schistosoma* parasites. Both *B. pfeifferi* and *Bulinus* spp. snails were found in the canals during the rice non-growing season, when the fields were predominantly growing onion, and recolonized the rice fields when they were re-filled for the next rice growing season. Although some species of *Bulinus* snails can also persist in the soil through droughts [28], the most likely source of snail recolonization to rice fields is from nearby irrigation canals with the reintroduction of water during the rice growing seasons.

Submerged vegetation had the most snails, followed by emergent vegetation and then open water. The observed association between snails and submerged vegetation is consistent with previous studies in the SRB that showed that submerged vegetation is strongly linked with both *B. pfeifferi* and *Bulinus* spp. abundance, and that these snails are rarely found in 100% coverage of emergent vegetation [29, 30]. Although our traps captured snails among the rice stalks, more thorough sampling would better identify the use of specific microhabitats by snails in these fields, which would allow for more targeted approaches and a greater understanding of the ecology of these snails in the fields. However, given that all the snail species showed an association with aquatic vegetation, periodic cleaning of vegetation from irrigation canals could provide a simple measure to reduce snail populations in both the canals and rice fields. This has been successful in other water bodies [29], as well as in irrigation canals in other parts of Africa [31].

This study fills an important gap in the literature by documenting *Schistosoma*-shedding snails in rice fields, verifying an occupational hazard for rice farmers. Previous studies considered occupational risk of schistosomiasis by examining for associations between area of rice cultivated and schistosomiasis infections in children [32, 33], general schistosomiasis prevalence [34], or by looking at the occupation of rice farming as a risk factor [35]. However, few studies have actually demonstrated the presence of *Schistosoma*-shedding snails in rice fields [36]. In our study, across fields and canals, the highest risk of schistosomiasis based on snail and cercarial abundance was during the rice harvest and growing season, a time when people are most likely to be in the rice field and canal water. Thus, our study demonstrates that there is an infection risk to rice farmers from field and adjacent canals, going beyond previous correlational studies that are plagued by third variable problems. Occupational risk of schistosomiasis for rice farmers in other parts of the world where

rain, spring, or swamp rice fields are employed might be different than the risk associated with irrigation-fed rice in the SRB, which warrants investigation. Our findings suggest that there would be value to snail control in rice fields, especially in areas where schistosomiasis transmission is not well controlled by traditional techniques, such as mass drug administration.

We explored one potential snail control method that could be effective in rice fields – the use of fish to reduce snail populations. Carp fish have been used in other areas to control snails, especially Apple snails in Asia; however, carp are non-native to Senegal and have been shown to harm native fish and ecosystems when introduced [21, 37–39]. Avoiding the use of an invasive fish is important to protect the greater SRB ecosystem and fishing livelihoods. While not yet optimized for Senegalese rice field cultivation within a schistosomiasis context, the four Senegal-native fish species we tested have been successfully used in rice aquaculture [16–18]. Furthermore, fish also have been shown to be beneficial to rice by reducing macroinvertebrate pests and increasing rice productivity [13, 14].

The value of fish predation might be diminished if Senegalese snails exhibit strong anti-predator behaviors in response to fish; however, in our study, native snails exhibited only moderate avoidance and refuge use responses to fish chemical cues. When vegetation was present, *Lymnaea* spp. were more likely to be at or above the water line than *B. pfeifferi* and *Bulinus globosus/truncatus* in response to chemical cues of fish relative to control water. Additionally, all three genera of snails showed a marginal increase in use of vegetation in the presence of chemical cues from some fish when compared to the water control. These results differ from Swartz et al. [38], which found predator avoidance behavior with snails exiting the water. However, they conducted their study with laboratory-raised intermediate host snails and crayfish predators [39], whereas we used wild-caught snails and fish. Additionally, Swartz et al. [40] used coupled crushed snails chemical cues with the predators cues, whereas we only used cues from predators that had not fed on snails. Furthermore, we also included vegetation (*C. demersum*) and no substrate, as *Biomphalaria* spp. and *Bulinus* spp. in the study area are found in *C. demersum* [29]. It is possible that the higher tendency of water quitting behavior seen in the lab snails of Swartz et al. [40] is not seen in the wild snails of our experiment because of the high risk of desiccation in the arid environment of Senegal, which is why we broadened our definition of avoidance behavior to near or above the water line. Although it is possible that snails require direct chemical cues from injured conspecifics to induce strong anti-predator responses and respond only weakly to fish chemical cues alone, the low but present usage of vegetation in tanks with fish chemical cues indicates that they do indeed respond to fish cues even if they have not been feeding on snails. Nevertheless, snail avoidance behaviors to the fish chemical cues detected by this study were not substantial and seem unlikely to completely prevent fish predation. Hence, we are hopeful that fish predation can reduce both snail and *Schistosoma* cercarial populations in rice fields.

Given that *Lymnaea* spp. snails are more likely to be found on emerged vegetation and were more likely to be above or at the water line in our experiment, the inability of fish to consume them could be a concern for the ability of rice-fish farming to reduce the incidence of fasciolosis. Some fish utilized in rice-fish co-culturing can reduce pests found on rice stalks above the water by knocking the pests into the water and then consuming them [41, 42], but we were unable to test this behavior with the four fish species tested. While fasciolosis is a major disease of economic and veterinary importance in Africa, only a few human fasciolosis cases have been reported in Senegal [43–45]. Local herders are aware of this liver fluke and reported seeing it upon slaughter or death of cattle and sheep. Further work on rice-fish farming for biocontrol should monitor *Lymnaea* spp. populations in rice fields in addition to *Bulinus* spp. and *Biomphalaria pfeifferi* populations.

Despite evidence of moderate antipredator behaviors in response to fish chemical cues, the predation experiments showed that *Hemichromis* spp. fish were successful predators of intermediate host snails and thus are a strong candidate for inclusion in an integrated rice-fish system. *Heterotis niloticus* are also a candidate for rice-fish co-culturing because they are a preferred fish for aquaculture according to local reports and were also observed consuming snails, although to a lesser extent than *Hemichromis*. All *H. niloticus* and *Hemichromis* spp. fish in this experiment were wild caught, and further experiments should be conducted to assess whether aquaculture-raised fish have different food preferences. While

not tested in this study, Nile tilapia (*O. niloticus*) will consume algae off of rice stalks [41], so their addition to fields could reduce algal resources to snail hosts if they are utilizing this emergent plant as microhabitat even though they did not consume snails. Thus, they too could reduce the risk of schistosomiasis by reducing snail densities or cercarial shedding by snails [9, 41]. Studies on snail-eating species often look at the presence of snail shells in stomach contents [17]. Our observations of *Hemichromis* spp. and *C. gariepinus* removing snails from their shells and other reports of this behavior in *Clarias* spp. suggest that the presence of snails shells is a problematic way to identify snail consumers [46].

Some fish exhibited gape-limited predation of snails. This was only the case for *Lymnea* and *Biomphalaria* species, as *Bulinus* were the smallest of the three snail species and were rarely too big to consume for even the smallest of the fish we tested. This is good news for schistosomiasis control because *S. haematobium* transmitted by *Bulinus* spp. is the most common cause of schistosomiasis in the region [29]. *Hemichromis* spp. was the one fish species that showed the strongest evidence of gape-limited predation because snail size was smaller before than after predation by *Hemichromis* spp. This indicates a greater risk of predation by *Hemichromis* for small than large snails. This was not surprising given that *Hemichromis* was the smallest fish species tested and thus was the most likely to be gape limited. Unfortunately, in field studies, larger snails are more likely to be infected and shedding schistosomes [9], which means that gape-limited predators could increase schistosomiasis risk by reducing competition from small snails and providing more food to large infected snails, thus increasing cercarial production. Picking fish predators that can consume all size snails, such as *H. niloticus*, for rice-fish co-culturing should reduce this risk.

Water temperatures were recorded as high as 30 °C during rice field sampling, and although *H. niloticus* and *Hemichromis* spp. fish are native to Senegal, natural water bodies may be cooler than the shallow rice fields. Therefore, experiments going forwards on rice-fish culturing should account for the high temperatures in the SRB, especially in the particularly arid Richard Toll region, ensuring that fish can survive if placed in rice fields. The use of a deeper canal area is typically added in rice-fish co-culturing to provide a refugia from the high temperatures of shallower waters and to mitigate desiccation risk if water levels drop [47]. To advance towards integrating biocontrol of schistosomiasis in rice fields in the SRB, next steps should include tests of the native fish species in rice fields to assess the feasibility, economic viability, and ecological and malacological effects of this system in a more complex, natural environment.

Overall, this work confirms an occupational risk of schistosomiasis to rice famers because snails shedding *Schistosoma* cercariae were regularly found in rice fields and neighboring irrigation canals across seasons and years, suggesting that snail control in rice fields might be worthwhile. However, we acknowledge that only morphological identification of *Schistosoma* cercariae species was conducted and thus it is possible that some of the *Schistosoma* were not cercariae that infect humans. Nevertheless, researchers using morphological identification are highly trained, and there is documented evidence that individuals working in rice fields urinate and defecate in and around rice fields [48], which would expose susceptible snails to human-*Schistosoma* species. Additionally, ongoing work seeks to construct direct linkages between rice fields and *Schistosoma* infection in rice farming communities, which will shed further light on the exposure risk to farmers. For the avoidance and predation experiment, we attempted to best replicate natural conditions by using wild caught, native snails and fish and plants directly collected from water sources. However, for the avoidance experiment, we only tested hourly, and some immediate, short-term avoidance behaviour may have been missed. For the predation experiment, we only provided snails as food and thus preferences for snails versus other food items remains equivocal. Ongoing work in actual rice fields will facilitate capturing these more complex and natural systems.

## Conclusion

Although praziquantel treatment of schistosomiasis can be effective, reinfection is extremely common. Previous studies reported proximity to irrigated agriculture, including rice fields, as a risk factor for schistosomiasis. Moreover, people from more rural and lower socioeconomic communities are at a higher risk of infection, perpetuating a negative feedback system between poverty and poor health [6]. A system of integrated rice-fish farming using fish that reduce snail populations and

*Schistosoma* transmission has the potential to target this poverty-disease trap [49]. Furthermore, integrated rice farming has the potential to reduce other inputs, such as fertilizer, herbicides, and insecticides, and the known health effects of these agrochemicals on ecosystems and communities [14]. Here, we established that rice fields present an exposure risk of schistosomiasis to rice farmers, that certain native Senegalese fish are predators of snails that spread human *Schistosoma* parasites, and that snail antipredator behaviors are not completely effective at preventing predation by these species. Thus, some native fish species, particularly *H. niloticus* and *Hemichromis* spp., are strong candidates for use as biocontrol in Senegalese rice fields because of their voracity on snails and use in rice-fish co-culturing and aquaculture in other parts of the world [18–21, 23]. Our study represents the first steps toward establishing rice-fish co-culturing in Senegal to reduce the occupational risk of schistosomiasis for rice farmers while simultaneously providing fish to support rice farming households.

## Supporting information

**S1 Fig. Average size of snails alive before and after exposure to fish predators, mean and standard deviation across snail species.** Only snails exposed to the smallest of the predators, *Hemichromis* spp., were larger after predation than before. ** = $p$-value ≤0.05 and ≥0.01.
(TIFF)

**S1 Table. Pairwise comparisons across fish species chemical cues by snail species and vegetation presence.** Tank and temporal block were included as random intercepts.
(DOCX)

## Acknowledgments

We would like to thank all the farmers and their families who allowed us to test in their fields and irrigation canals and kindly shared their time with us. We would also like to thank everyone who worked at the Station d'Innovation Aquacole for your support and patience, especially during the aquaculture experiments. This work could not have happened without you all.

## Author contributions

**Conceptualization:** Alexandra Sack, Emily Selland, Jason R. Rohr.

**Data curation:** Alexandra Sack, Emily Selland.

**Formal analysis:** Alexandra Sack, Emily Selland.

**Funding acquisition:** Emily Selland, Jason R. Rohr.

**Investigation:** Alexandra Sack, Emily Selland, Sidy Bakhoum, Momy Seck, Nicolas Jouanard, Louis Dossou Magblenou.

**Methodology:** Alexandra Sack, Emily Selland, Jason R. Rohr.

**Project administration:** Momy Seck, Jason R. Rohr.

**Resources:** Momy Seck, Nicolas Jouanard, Louis Dossou Magblenou.

**Supervision:** Jason R. Rohr.

**Writing – original draft:** Alexandra Sack, Emily Selland.

**Writing – review & editing:** Alexandra Sack, Emily Selland, Sidy Bakhoum, Momy Seck, Nicolas Jouanard, Louis Dossou Magblenou, Jason R. Rohr.

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
