## [Decision Letter · Decision Letter 0]

PGPH-D-25-00328

Human Schistosoma exposure risk in rice fields and an exploration of fish species for snail and schistosomiasis biocontrol

Dear Dr. Croft,

Thank you for submitting your manuscript to PLOS Global Public Health. After careful consideration, we feel that it has merit but does not fully meet PLOS Global Public Health’s publication criteria as it currently stands. Therefore, we invite you to submit a revised version of the manuscript that addresses the points raised during the review process.

In addition to providing a point-by-point response to the reviewer comments, please ensure that you have followed the PLOS Data Policy regarding data availability. 

We look forward to receiving your revised manuscript.

Kind regards,

Mara Jana Broadhurst, M.D., Ph.D.

Academic Editor

Journal Requirements:

1. In the online submission form, you indicated that [Data and R code is available upon request from the corresponding author.].

a. In a public repository,

b. Within the manuscript itself, or

c. Uploaded as supplementary information.

Additional Editor Comments (if provided):

Reviewers' comments:

Reviewer's Responses to Questions

**Comments to the Author**

1. Does this manuscript meet PLOS Global Public Health’s publication criteria?

Reviewer #1: Yes

Reviewer #2: Partly

2. Has the statistical analysis been performed appropriately and rigorously?

Reviewer #1: Yes

Reviewer #2: Yes

3. Have the authors made all data underlying the findings in their manuscript fully available (please refer to the Data Availability Statement at the start of the manuscript PDF file)?

Reviewer #1: Yes

Reviewer #2: No

4. Is the manuscript presented in an intelligible fashion and written in standard English?

Reviewer #1: Yes

Reviewer #2: No

Reviewer #1: The authors are to be commended for an interesting, thoughtful and scholarly manuscript that brings data on schistosome and molluscan host distribution together with novel information on interactions between snails and potential fish predators. I have no major concerns with the methodology, analysis of the data, and conclusions reached, although the manuscript would be improved if some minor issues are addressed:

I am curious whether there is any data on the seasonality of human schistosome infection in the area, and whether this correlates with rice farming activities.

As snail-fish trophic interactions are a major theme of the manuscript, it would benefit from at least some background information on how snails detect predatory fish and what avoidance behaviors are employed to avoid predation. Is there anything known of the molecular nature of fish signals and how snails detect these signals?

Line 126 – how confident are the authors in their identification of cercariae by microscopy alone? Were any molecular analyses employed to confirm species identification? Numerous Schistosoma species are found in West Africa – several species of veterinary importance in addition to the human pathogens are found in Senegal and neighboring countries, and hybridization between some species has been well documented.

Line 123 – “Bulinus foskalii” should be “Bulinus forskalii”. Also see labeling of Figure 1.

Line 212 – define “AICc” (Akaike information criterion)

Labeling in the figures is small and hard to read, even when viewing the downloaded hi-res TIFF files. Suggest redoing all the labeling in a larger font.

Reviewer #2: • The rationale and knowledge gap for the study should be more clearly articulated.

• A section outlining the overall study design is currently missing and should be included.

• A data analysis section summarizing the statistical methods used is needed. The models described for each experimental procedures could be included in this section.

• The description of the experimental approach could be more concise by referencing prior studies that detail predator avoidance and selective predation procedures.

• The discussion section lacks focus and depth in some areas, which weakens the overall argument of the paper.

• Including a paragraph that summarizes the strengths and limitations of the study would be valuable.

• Significant formatting revisions are required to meet publication standards, particularly regarding font size, spacing, and table presentation.

• Substantial revisions to the manuscript’s language are needed to improve clarity, coherence, and overall flow.

**Do you want your identity to be public for this peer review?** For information about this choice, including consent withdrawal, please see our Privacy Policy

Reviewer #1: **Yes: ** Stephen J. Davies

Reviewer #2: No

---

## [Editor Report · Decision Letter 1]

Human Schistosoma exposure risk in rice fields and an exploration of fish species for snail and schistosomiasis biocontrol

PGPH-D-25-00328R1

Dear Dr. Rohr,

We are pleased to inform you that your manuscript 'Human Schistosoma exposure risk in rice fields and an exploration of fish species for snail and schistosomiasis biocontrol' has been provisionally accepted for publication in PLOS Global Public Health.

Best regards,

Mara Jana Broadhurst, M.D., Ph.D.

Academic Editor
